# How a Rubric Score Application Empowers Teachers' Attitudes over Computational Thinking Leverage

Ioannis Dimos [1,*], Chrysoula Velaora [1], Konstantinos Louvaris [2], Athanasios Kakarountas [1] and Assimina Antonarakou [2]

1   Department of Computer Science and Biomedical Informatics, University of Thessaly, 35101 Lamia, Greece
2   Didactics of Geosciences, National and Kapodistrian University of Athens, 15772 Athens, Greece
*   Correspondence: ioadimos@uth.gr

**Abstract:** Computational Thinking (CT) has emerged as an umbrella term that refers to a broad set of problem-solving skills. New generations must conquer these skills in order to thrive in a computer-based world. Teachers, as agents of change, must also be familiar, trained and well-prepared in order to train children in CT. This paper examines STEM (Science, Technology, Engineering and Mathematics) and non-STEM teachers' attitudes and readiness to adopt and utilize Computational Thinking concepts in the curriculum. The research was conducted through a descriptive assessment of students using thematically related criteria (rubrics) and a criterion on Computational Thinking usage and utilization. Fifteen teachers (n = 15) were invited to a focus group discussion in which they were asked to complete a questionnaire and, subsequently, to openly analyze their answers. The results show that the majority of teachers used computational thinking as an assessment criterion and stated that they did not face any significant problems with it. At the end of the focus group questions, they concluded that they consider participation in a training program regarding the concept and principles of computational thinking and the way they could integrate into the educational process necessary. Teachers expressed their confidence in using a set of criteria (rubric) to make students' assessments more effective and stated that they can easily use at least one criterion for Computational Thinking.

**Keywords:** computational thinking; descriptive assessment; rubric; application; student's performance; student's competences

## 1. Introduction

Computational Thinking (CT) is undoubtedly considered a fundamental skill for reading, writing and arithmetic in the 21st century [1]. Improving her initial definition of CT [2], Janette Wing expressed that Computational Thinking refers to the mental processes involved in formulating a problem and expressing its solution(s) in such a way that a computer—human or machine—can carry out the task effectively [3]. She declared that everyone can benefit from thinking computationally [1]. Even though the CT definition was a subject of discussion for many years by the scientific community [4,5], eventually, CT became commonly accepted as a *problem-solving method* [6]. Beyond that general definition, teachers or students who are dealing with CT teaching and/or training mainly develop skills such as *abstraction, decomposition, generalisation, algorithmic thinking, and evaluation* [6,7].

Problem-solving skills encapsulate the ability to think, otherwise known as critical thinking. Critical thinking is best understood as the ability of thinkers to take charge of their own thinking [8]. It is the art of analysing and evaluating thinking with a view toward improving it [9]. Thinkers improve their quality of thinking with sound criteria and standards that they have developed [8]. Minds that are flexible, adaptable and experienced in constantly thinking and rethinking issues are ready to face the demands of the 21st century [8].

Currently, there are plenty of learning and teaching materials related to CT that propose activities aimed at developing CT competencies and are available in various formats.

These materials can be utilised by both STEM and non-STEM teachers. An example of this type of educational material is available in the project CS Unplugged [10], which explains how the concepts of CT (algorithmic thinking, abstraction, decomposition, generalisation and patterns, logic and evaluation) can be applied to each pedagogical activity. Although there are different approaches [11–13] that relate the proposed educational activities to the development of CT skills, there are no instruments that, quantitatively and qualitatively, aim to depict and evaluate teachers' attitudes and readiness to adopt and utilise CT concepts in the curriculum based on the students' assessment approach.

Thus, in the present work, we focused on students' assessment procedures by adopting the Descriptive Assessment approach and creating assessment criteria (Rubrics) related to the teaching topic with the addition of one CT criterion. These assessment criteria were applied to students by using our own Rubrics Score Application. At the end of the teachers' assessment period, the researchers analysed the collected data and concluded specific findings. The researchers also organised a participatory focus group and, in conjunction with the assessment data, formulated the research results.

## 2. Qualitative Assessment of Students' Performance

There are many platforms available on the internet that can record assessment results using Rubrics, such as Rubistar or For All Rubrics. All of these platforms have a major disadvantage—they only report summative assessment results without analysing them in order to provide a formative direction to the assessment. In this context, a Rubric Score Application was developed by the researchers to analyse assessment results in real-time at whatever level and form the teacher chooses (numerically or descriptively or in both ways). This application automatically generates a Descriptive Assessment Report for students' performance and was implemented by the researchers in order to assess students' performance in Computational Thinking. To the default analytical rubric of the application, the researchers added one more criterion. Assessment as a term refers to the judgement of students' performance in relation to specific goals, and a formative direction requires (a) feedback and (b) an indication of how the work can be improved to reach the required standard [14–17]. Using the term *"feedback"* we adopted the definition of Ramaprasad [18,19], who describes feedback as the distance between the actual and reference levels of the system parameters, and this is used to alter the gap in some way.

The starting point of this research is to enable teachers to improve their teaching and students to improve their learning by changing aspects of the assessment methodology toward a formative and descriptive direction. Usually when teachers inform parents aurally or with assessment's notes, they use stereotyped expressions such as "he is good..." or "more effort is needed...", without focusing on learning results in terms of cognitive or other goals. The descriptive form of reporting assessment results is ideal for the provision of clear and analytical information. Generally, two types of rubrics (table of criteria) can be used for educational purposes. The first is an "analytic rubric". For this type, each dimension or criterion is evaluated separately, and a student assessment that provides information on criteria about the student's weak and robust points for each task. This information could be used for future student improvement. The second type is a "holistic rubric" in which all dimensions are assessed simultaneously to provide a single overall score. Analytic rubrics requires more time to score tasks in comparison with holistic rubrics, but it is suitable for formative assessments [20]. Most educators recommend the use of analytic rubrics for effective assessments. A rubric provides three essential features: evaluative criteria, quality definitions for those criteria at particular quantitative levels of performance, and a scoring strategy [21,22]. The use of rubrics in the educational process offers many benefits to students and teachers. Rubrics (a) inform students of expectations, (b) provide feedback, (c) maintain consistent grading and fair assessment, and (d) enhance student learning and self-assessment [20]. In addition, rubrics provide teachers with mechanisms to (a) clarify teaching and learning goals, (b) analyse student scores with specific criteria and skills, (c) summarise student performance reliably, and (d) identify patterns of strengths

and weaknesses in students' work [23]. The words used to identify quality levels may come from the learning outcomes in Bloom's taxonomy, which provides a vocabulary for increasingly complex levels of instructional goals [24]. Through the use of analytic, topic-specific rubrics, the reliability of scoring of performance assessments can be enhanced, especially if they are complemented with the training of raters [25]. The Rubric Score Application developed by the researchers is described in paragraph V.

## 3. Related Work

Descriptive assessment of students in K-12 classrooms is supported by the use of rubrics. When teachers use them, student outcomes, learning engagement, learning skills and teacher practice are enhanced [26]. Researchers used a rubric to assess student (a) learning [27,28] and computational thinking [29] in programming environments and (b) critical thinking, information processing [30] and computational thinking performance [31] in STEM courses. Researchers have proposed methods for assessing the learning outcomes of CT skills that are applicable in programming environments. When comparing published information on rubrics that exist in higher education and K-12, it can be observed that higher education users are more likely to publish their results [32]. Analytical and descriptive rubrics are mainly studied in higher education and consist of four or five levels of performance, and all previous studies describe positive outcomes for the use of rubrics [32]. E-rubric can be an effective evaluation tool for education that can improve the quality of the students' skills and the quality of the learning process [33]. Teachers should consider the use of rubrics for evaluation to motivate students to use them for self-evaluation [34]. Moreover, as far as the online learning assessment is concerned, a measurement model based on metrics in combination with techniques such as 'peer interaction', 'forum activities', 'learning by doing', and 'systematic feedback' is used. This model enables teachers to measure students' performance and develop a more adaptive teaching approach [35].

## 4. Technology Acceptance Model (TAM)

Davis [36] developed the Technology Acceptance Model (TAM), which theorises that an easy-to-use and useful technology will have a positive influence on the user's attitude and intention to use it. Davis [36] defined two terms, the perceived usefulness (PU) and the perceived ease-of-use (PEOU). The former describes the extent to which a person believes that using a specific system will improve his or her job performance, while the second is the degree to which a person believes that using the system will be easy. Perceptions of usefulness were found to be stronger and consistent with the acceptance of information technology compared to other variables, such as attitudes, satisfaction and other perceptual measures [37,38]. The present study used the Technology Acceptance Model (TAM) to investigate the effects of TAM variables on teachers' satisfaction regarding the Rubric Score Application. The conceptual framework model consisted of perceived ease-of-use (PEOU) variables, perceived usefulness (PU) variables and satisfaction (SAT) variables.

Previous studies demonstrated the effects of perceived usefulness and perceived ease-of-use on students' [39,40] and teachers' [41] satisfaction regarding e-learning systems. Researchers used the TAM model to evaluate the acceptance of the use of virtual platforms by university students [42]. Based on the literature review, a set of research questions is proposed:

RQ1. Is it probable that this Rubric Score Application will be perceived as useful to a significant extent by the teachers?

RQ2. Is it probable that this Rubric Score Application will be perceived as easy to use to a significant extent by the teachers?

RQ3. Does the perceived ease-of-use of this Rubric Score Application have a significant effect on its perceived usefulness?

RQ4. Do the perceived usefulness and perceived ease-of-use significantly affect teachers' satisfaction?

RQ5. Is there a significant difference between teachers who teach STEM and non-STEM courses in terms of the frequency of usage of computational thinking as an assessment criterion?

RQ6. What are the teachers' attitudes towards the use of computational thinking as an assessment criterion?

## 5. Descriptive Assessment Application

A descriptive Rubric Score Application was developed for use in the educational process in the area of student performance assessment by primary and secondary teachers. The application analyses the assessment results at whatever level the teacher chooses (students, department, class, school, etc.) in real time and expresses them numerically or descriptively or in both ways depending on the choice of the teacher, as shown in Figure 1 (the text is in the Greek language). One of the rubric applications is presented in Table A1 (see Appendix A). It consists of six evaluative criteria with Graded Criteria Scales and evaluates students across six dimensions: reading comprehension, writing, critical thinking, participation—collaboration, diligence and computational thinking. The teacher has the ability to assess the student on any of these criteria. All criteria are scored on a 4-point scale, with 1 representing the minimum score and 4 representing the maximum score, as shown in Figure 2. Each score contributes to the evaluation result with a certain coefficient.

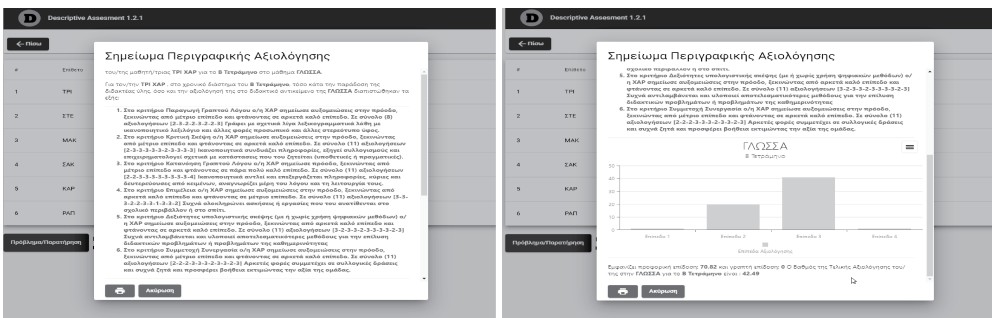

**Figure 1.** Assessment results (with and without graph).

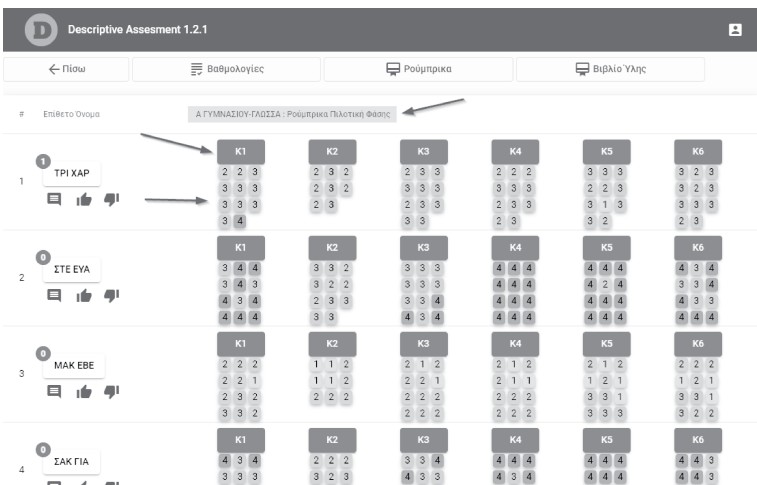

**Figure 2.** Scorecard.

All evaluative data are stored in a remote database. The application is online, and the teachers' connected to it with the credentials obtained from the researchers. The means for using the application and recording the evaluation results can be a tablet, mobile phone, laptop or desktop computer.

## 6. Material and Methods

### 6.1. Research Population

This study included 15 primary and secondary education teachers and was conducted during the second quarter of the 2020–2021 school year in Greece. The research sample was selected by convenience sampling and can be considered satisfactory. However, it limits the generalisation of the findings, so they cannot be applied to a broader context. This number of participants was selected in accordance with Sevilla et al. (1984) [43], who stated that a minimum sample size for experimental research is 15 people [44,45]. The pandemic situation discouraged teachers from participating in the research phase, creating a barrier to the generalisation of the research results. Of these teachers, 11 were female and 4 were male, 3 were primary school teachers and 12 were secondary schoolteachers. In terms of experience in descriptive assessment, 7 teachers declared that they had no or low experience, and 8 teachers declared that they had satisfactory or good experience. Finally, 8 teachers declared that they had no or low experience in using ICT for student assessments, and 7 teachers stated that they had satisfactory or good experience. Additionally, teachers were asked about their beliefs on computational thinking. For this, 14 teachers agreed that they could teach their course using CT principles and 1 teacher had a neutral view. Thirteen teachers agreed that they could teach CT principles through their course, 1 teacher had a neutral view, and 1 teacher did not answer the question. Seven teachers agreed that CT should be taught as a separate course, 7 teachers had a neutral view, and 1 teacher disagreed with this statement. The participant background information is presented in Table 1.

**Table 1.** Participant background information.

| Participant Background Information | N | Percentages |
|:---:|:---:|:---:|
| **Gender** | | |
| Male | 4 | 26.7% |
| Female | 11 | 73.3% |
| **Grade level taught** | | |
| Primary | 3 | 20.0% |
| Secondary | 12 | 80.0% |
| **Experience** | | |
| **Descriptive student assessment** | | |
| Not at all, Low | 7 | 46.7% |
| Satisfactory, Good | 8 | 53.3% |
| **ICT student assessment** | | |
| Not at all, Low | 8 | 53.3% |
| Satisfactory, Good | 7 | 46.7% |
| **Teaching using CT principles** | | |
| Agree | 14 | 93.3% |
| Neutral | 1 | 6.7% |
| **Teaching CT principles** | | |
| Agree | 13 | 86.6% |
| Neutral | 1 | 6.7% |
| No answer | 1 | 6.7% |
| **Teaching CT as a separate course** | | |
| Agree | 7 | 46.7% |
| Neutral | 7 | 46.7% |
| Disagree | 1 | 6.6% |

### 6.2. Data Collection Methods

To evaluate our Rubric Score Application, a concurrent triangulation strategy with mixed methods was used. This is an approach in which the researcher collects quantitative and qualitative data concurrently. Then, the two databases are compared to determine

whether there is convergence, differences, or some combination of the two [46]. Our goal for using multiple methods was to find the most accurate and valid answers to our research questions. One strategy used by Ref. [47] to strengthen the reliability as well as the internal validity is triangulation. Thus, quantitative research with a structured questionnaire and qualitative research with a focus group method were used. Focus group methods can be used to conduct action research. As the research participants can become an active part of the analysis process, focus group methods can empower them [48]. Questionnaires are more suitable for obtaining specific quantitative information that depicts the number of people and expresses a predefined opinion. Focus groups are appropriate for exploring exactly how those opinions are constructed [48].

All teachers (N = 15) received an online questionnaire based on the TAM using Google forms, and 15 responded positively. Some modifications were made for the purpose of this study, and four factors were included in the questionnaire: perceived usefulness (PU), perceived ease-of-use (PEOU), satisfaction (SAT) and computational thinking (CT). There were 9 items (Q1–Q9), for which 5 Likert-type scale answers ranging from 1 to 5 (1 = Strongly Agree, 2 = Agree, 3 = neither disagree or agree, 4 = Disagree or 5 = Strongly Disagree); and 2 items (Q10–Q11) were adopted and 4 Likert-type scale answers ranging from 1 to 4 (1 = Not at all, 2 = Little Extent, 3 = Some Extent, 4 = Great Extent) were adopted. The items included in the questionnaire are presented in Table 2.

All teachers (N = 15) received an online invitation by e-mail to participate in focus group research, and 12 responded positively. Finally, 2 focus groups were formed, one in Lamia and one in Athens. The Lamia focus group (Focus Group A) included 5 teachers, 1 male and 4 females. The Athens focus group (Focus Group B) included 7 teachers, 1 male and 6 females. It is accepted that between five and ten participants in a focus group research study is sufficient [49].

**Table 2.** Online questionnaire.

| Factors | Items | Questions |
|---|---|---|
| **PU** | | The use of this application helped me to . . . |
| | Q1 | become more productive. |
| | Q2 | communicate essentially with parents. |
| | Q3 | study the data and differentiate my teaching. |
| | Q4 | monitor and evaluate the material I have taught. |
| **PEOU** | | I believe that the assessment application that I used. . . |
| | Q5 | works as I expected it to. |
| | Q6 | is stable during its operation. |
| | Q7 | is compatible with school procedures. |
| **SAT** | Q8 | facilitates the teachers with the assessment of the students during distance learning. |
| | Q9 | if it was commercially available (at an affordable price) I think it is a worthwhile purchase for teachers. |
| **CT** | Q10 | To what extent did you engage with the computational thinking criterion? |
| | Q11 | To what extent did the computational thinking criterion match your subject? |

### 6.3. Data Analysis Methods

The primary method used for the data analysis involved quantitative data analysis techniques, which are described below. The IBM SPSS 26 statistical package was used to analyse the data.

#### 6.3.1. Validity Test

The validity was analysed by using the Spearman rank correlation coefficient (rho) to determine the strength of the relationships among the question items. The items on the questionnaire were considered valid if the rcount of the results was greater than the rtable. The appearance of a strong correlation coefficient indicates that the measurement tool used is valid [50].

#### 6.3.2. Reliability Test

The reliability was analysed using Alpha Cronbach statistics as a measurement to determine the consistency of respondents' answers with the question items. When measuring reliability using Alpha Cronbach statistics, a variable can be considered reliable if it has an alpha value greater than 0.60.

#### 6.3.3. Normality Test

The normality test was used to determine whether the data used in the research were normally distributed or not. One reliable method is to determine the normal probability plot which compares the cumulative and the normal distribution. This can show whether the data spread around the diagonal line and follow the direction of the diagonal line. A second reliable method is to use the Shapiro–Wilk test. If the significance value is greater than 0.05, the data are considered normally distributed.

#### 6.3.4. One Sample *T*-Test

The one-sample *t*-test was used to specify the case of one concrete value being differentiated from the mean of an unspecified population. A significance value of less than 0.05 indicates that then the population mean is not different from the specific value.

#### 6.3.5. Randomness Test

The runs test was used to determine whether a data set is from a random process. A significance value of less than 0.05 indicates that the sequence of observations was produced in a random manner, that is, the runs test ensures the independence of the data.

#### 6.3.6. Homoscedasticity Test

Levene's test was used to evaluate the homogeneity assumption, that is, whether the dependent variable has the same variance across different groups. Variance is the expectation of the squared deviation of a random variable based on its population mean or sample mean. A significance value of less than 0.05 indicates that the groups being compared have equal variance, that is, the sample observations have homoscedasticity.

#### 6.3.7. Multiple Linear Regression Analysis

A simple linear regression is used if one independent variable is included in the regression model. A multiple linear regression is used if two or more independent variables are included in the regression model. The linear regression analyses the effects of the independent variables on the dependent variable. The sample observations must (a) follow a normal distribution; (b) be simple random samples, which means that the samples must be independent of each other; and (c) have homoscedasticity or homogeneity of variance, which means that variances of the samples must be equal.

### 6.3.8. *T* Test

The T statistical test is used to test the success of the regression coefficient impartially. significance value of less than 0.05 indicates that the independent variable has a significant effect on the dependent variable.

### 6.3.9. F Test

The F statistical test was used to simultaneously determine the effect of the independent variables on the dependent variable. A significance value of less than 0.05 indicates that the independent variables have a significant simultaneous effect on the dependent variable.

### 6.3.10. The Coefficient of Determination (R Square)

The coefficient of determination (R Square) was used to measure the level of the model's ability to explain the dependent variable.

### 6.3.11. Mann–Whitney U Test

The Mann–Whitney U test is a nonparametric test that is used to determine the equality of means in two independent samples that come from the same population. This test has the great advantage of being appropriate for small samples (5 to 20 participants) [51]. A significance value of less than 0.05 indicates that there are no differences between the two samples, and at a significance level of 0.05, the null hypothesis is accepted.

## 7. Results

The results of the quantitative and qualitative data analyses are presented below.

### 7.1. Validity Test

Table 3 shows the rcount results, which were compared with the rtable value. The r-table value was obtained from the r-table distribution with a significant level of 0.05 using a two-sided test (n = 15). As the rcount values of the perceived usefulness, perceived ease-of-use, satisfaction and computational thinking variables were greater than the rtable value, all research instruments were deemed to be valid.

**Table 3.** Validity of the items.

| Factors | Items | r (Item, Total) |
|---|---|---|
| **PU** | Q1 | 0.820 |
| | Q2 | 0.641 |
| | Q3 | 0.749 |
| | Q4 | 0.555 |
| **PEOU** | Q5 | 0.794 |
| | Q6 | 0.913 |
| | Q7 | 0.903 |
| **SAT** | Q8 | 0.863 |
| | Q9 | 0.888 |
| **CT** | Q10 | 0.823 |
| | Q11 | 0.850 |

### 7.2. Reliability Test

Table 4 shows the reliability test results for the perceived usefulness, perceived ease-of-use, satisfaction and computational thinking variables. The internal-consistency coefficient showed a Cronbach's alpha value of 0.859, indicating good reliability among the questionnaire items. The internal-consistency coefficient of all factors used in this study showed Cronbach's alpha values ranging from 0.554 to 0.831, so all factors are considered reliable.

**Table 4.** Reliability of the factors.

| Factors | Items | Cronbach's Alpha |
|---|---|---|
| PU | 4 | 0.624 |
| PEOU | 3 | 0.831 |
| SAT | 2 | 0.708 |
| CT | 2 | 0.554 |

### 7.3. Correlation Analysis

Through the correlation coefficients, the relationships among the three factors were discovered, and the hypotheses of the research model were investigated.

Table 5 shows that the correlation between the perceived ease-of-use and perceived usefulness variables and that between the perceived usefulness and satisfaction variables were positive and significant at the 0.01 level (2-tailed). This confirms the original hypothesis made in the literature concerning the Technology Acceptance Model (TAM).

**Table 5.** Correlations of factors.

| Factors | PU | PEOU | SAT | CT |
|---|---|---|---|---|
| PU | 1.000 | 0.700 | 0.815 | 0.365 |
| PEOU | 0.700 | 1.000 | 0.482 | 0.082 |
| SAT | 0.815 | 0.482 | 1.000 | 0.354 |
| CT | 0.365 | 0.082 | 0.354 | 1.000 |

### 7.4. Research Question 1

The first above-mentioned study research question was tested through the following hypotheses:

- H0: It is probable that the Rubric Score Application will be perceived as useful to a significant extent by the teachers. (The perceived usefulness is statistically equal to a value of 2.)
- H1: It is not probable that the Rubric Score Application will be perceived as useful to a significant extent by the teachers. (The perceived usefulness is not statistically equal to a value of 2.)

#### 7.4.1. Normality Test

Using the Shapiro–Wilk normality test, we concluded that the significance value of the perceived usefulness (0.495) is greater than 0.05, so the data are normally distributed.

#### 7.4.2. One Sample *T*-Test

Using the one-sample *t*-test, we concluded that the significance value of the perceived usefulness (0.830) is greater than 0.05, so the H0 hypothesis was accepted and the sample mean was not different from 2. Thus, it is probable that this platform will be perceived as useful to a significant extent by the teachers.

### 7.5. Research Question 2

The second above-mentioned study research question was tested through the following hypotheses:

- H2: It is probable that the Rubric Score Application will be perceived as easy to use to a significant extent by the teachers. (The perceived ease-of-use is statistically equal to a value of 2.)
- H3: It is not probable that the Rubric Score Application will be perceived as easy to use to a significant extent by the teachers. (The perceived ease-of-use is not statistically equal to a value of 2.)

### 7.5.1. Normality Test

Using the Shapiro–Wilk normality test, we concluded that the significance value of the perceived ease-of-use (0.258) is greater than 0.05, so the data are normally distributed.

### 7.5.2. One Sample *T*-Test

Using the one-sample *t*-test, we concluded that the significance value of the perceived ease-of-use (0.363) is greater than 0.05, so the H2 hypothesis was accepted and the sample mean was not different from 2. Thus, it is probable that this platform will be perceived as easy to use to a significant extent by the teachers.

### *7.6. Research Question 3*

The third above-mentioned study research question was tested through the following hypotheses:

- H4: The perceived ease of use does not have a significant effect on the perceived usefulness.
- H5: The perceived ease of use have a significant effect on the perceived usefulness.

### 7.6.1. Normality Test

Using the Shapiro–Wilk test on the residual values, the significance value was found to be 0.939, and the residual values of the perceived ease-of-use variable towards the perceived usefulness value were normally distributed.

### 7.6.2. Randomness Test

Using the Runs test on the residual values, the significance value was found to be 0.102, and the residual values of the perceived ease-of-use variable towards the perceived usefulness variable were independent.

### 7.6.3. Homoscedasticity Test

Using Levene's test on the residual values, the significance value based on the median was found to be 0.165, and the residual values of the perceived ease-of-use variable towards the perceived usefulness variable were homoscedastic.

### 7.6.4. Simple Linear Regression Analysis

A simple regression analysis can be formulated with the following formula:

$$PU = 0.762 + 0.653 \, PEOU + error$$

### 7.6.5. Coefficient of Determination

The regression constant was found to be 0.762, which means that if the perceived ease-of-use variable has a value of 0, then the value of the perceived usefulness variable will be 0.762.

The regression coefficient value of the perceived ease-of-use variable was found to be 0.653, which is positive. This means that if the value of the perceived ease-of-use variable increases by one unit, the value of the perceived usefulness variable will increase by 0.653 units.

### *7.6.6. T Test*

Comparing the perceived ease-of-use variable to the perceived usefulness variable produced a T significance value of 0.003, which is less than 0.05. Thus, the H4 hypothesis was rejected. These results conclude that the perceived ease-of-use variable has a significant effect on the perceived usefulness.

### 7.6.7. F Test

Comparing the perceived ease-of-use variable to the perceived usefulness variable produced an F significance value of 0.003, which is less than 0.05. These results allowed is to conclude that the perceived ease-of-use variable has a significant effect on the perceived usefulness.

### 7.6.8. The Coefficient of Determination (R Square)

The R square value was found to be 0.505, and this indicates that the perceived ease-of-use variable has an effect proportion of 50% towards the perceived usefulness variable, while the remaining 50% (100–50%) is influenced by other variables that were not examined in this research.

### 7.7. Research Question 4

The fourth above-mentioned study research question was tested through the following hypotheses:

- H6: The perceived usefulness and perceived ease-of-use do not have significant effects on satisfaction.
- H7: The perceived usefulness and perceived ease-of-use have significant effects on satisfaction.

### 7.7.1. Normality Test

Using the Shapiro–Wilk test on the residual values, we found a significance value of 0.677, and the residual values of the perceived ease-of-use and perceived usefulness variables towards the satisfaction variable were normally distributed.

### 7.7.2. Randomness Test

Using the Runs test on the residual values, the significant value was determined to be 0.986, and the residual values of the perceived ease-of-use and perceived usefulness variables towards the satisfaction variable were independent.

### 7.7.3. Homoscedasticity Test

Using the Levene's test on the residual values, the significant value based on the median was determined to be 0.404, and the residual values of the perceived ease-of-use and perceived usefulness variables towards the satisfaction variable were homoscedastic.

### 7.7.4. Linear Regression Analysis

A regression analysis can be formulated with the following formula:

$$\text{SAT} = -0.085 + 0.925\text{PU} + \text{error}$$

### 7.7.5. Coefficient of Determination

The regression constant was determined to be $-0.085$, which means that if the perceived usefulness variable has a value of 0, the satisfaction variable will be $-0.085$.

The regression coefficient value of the perceived usefulness variable was found to be 0.925, which is positive. This means that if the perceived usefulness variable increases by one unit, the satisfaction variable will increase by 0.925 units.

The perceived ease-of-use variable was excluded from the formula, as shown by Spearman's correlation.

### 7.7.6. T Test

Comparing the perceived usefulness variable to the satisfaction variable generated a T significance value of 0.000, which is less than 0.05, so the H6 hypothesis was rejected. These results allowed us to conclude that the perceived usefulness variable has a significant effect on the satisfaction variable.

### 7.7.7. F Test

Comparing the perceived usefulness variable to the satisfaction variable generated an F significance value of 0.000, which is less than 0.05. These results allowed us to conclude that the perceived usefulness variable has a significant effect on the satisfaction variable.

### 7.7.8. The Coefficient of Determination (R Square)

The R square value was found to be 0.700, and this indicates that the perceived usefulness variable has an effect proportion of 70% towards the satisfaction variable, while the remaining 30% (100–70%) is influenced by other variables that were not examined in this research.

### 7.8. Research Question 5

The fifth above-mentioned study research question was tested through the following hypotheses:

- H8: There is no significant difference between STEM and non-STEM teachers in terms of the frequency of use of the computational thinking criterion.
- H9: There is a significant difference between STEM and non-STEM teachers in terms of the frequency of use of the computational thinking criterion.

The subcategories of the CT criterion that were found to be used by teachers are:

1. The ability to describe and represent a problem (Abstractive/Algorithmic thinking) (C1)
2. Computational thinking skills (with or without the use of digital methods) (C2)
3. The emergence of Scientific Practice Skills (observing, recording, classifying, comparing) (C3)
4. Computational thinking skills (Designing and solving problems using programming techniques) (C4)
5. Critical Thinking (C5)

In order to test the equality of means in teachers who were teaching STEM and non-STEM courses, the Mann–Whitney U test was used. The term "STEM education" refers to teaching and learning in the fields of science, technology, engineering and mathematics [52]. Teacher T8 was excluded because he taught both STEM and non-STEM courses. The data on the frequency of use of the CT Criteria by teachers from STEM and Non-STEM courses, which were collected from the database, are presented in Tables 6 and 7. The significance value (0.482) is greater than 0.05, so the H8 hypothesis was accepted. These results allowed us to conclude that there is no significant difference between STEM and non-STEM teachers in terms of the frequency of use of the computational thinking criterion.

**Table 6.** Frequency of use of the CT Criteria by teachers in STEM courses.

| Teachers | Frequency | | | | | Sum |
|---|---|---|---|---|---|---|
| | **C1** | **C2** | **C3** | **C4** | **C5** | |
| T1 | - | 12 | 31 | - | - | 43 |
| T2 | - | - | 13 | - | - | 13 |
| T3 | - | 2 | - | - | - | 2 |
| T4 | - | 7 | 14 | - | - | 21 |
| T5 | 77 | 37 | - | 33 | - | 147 |
| T6 | 14 | 13 | - | - | - | 27 |
| T7 | 54 | 48 | - | - | - | 102 |
| T8 | - | 38 | 20 | - | - | 58 |

### 7.9. Research Question 6

The potential hypotheses regarding conflict are:

- H10: The teachers' attitudes towards computational thinking as an assessment criterion are positive. (The variable computational thinking is statistically equal to a value of 3.)
- H11: The teachers' attitudes towards computational thinking as an assessment criterion are not positive. (The variable computational thinking is not statistically equal to a value of 3.)

As shown in Figure 3, the data spread around the diagonal line, follow the direction of the diagonal line, and are normally distributed. Using the one-sample *t*-test, it can be concluded that the significance value of computational thinking (0.265) is greater than 0.05, so the H10 hypothesis was accepted and the sample mean was not different from 3. Thus, the teachers used the criterion of computational thinking without facing great difficulties regarding the teaching of their subjects.

**Table 7.** Frequency of use of the CT Criteria by teachers in Non-STEM courses.

| Teachers | Frequency | | | | | Sum |
|---|---|---|---|---|---|---|
| | **C1** | **C2** | **C3** | **C4** | **C5** | |
| T8 | - | 53 | - | - | 19 | 72 |
| T9 | - | 23 | - | - | - | 23 |
| T10 | 29 | 24 | - | - | - | 53 |
| T11 | - | 71 | - | - | 71 | 142 |
| T12 | - | 3 | - | - | - | 3 |
| T13 | - | 35 | - | - | - | 35 |
| T14 | - | - | - | - | - | 0 |
| T15 | - | - | - | - | - | 0 |

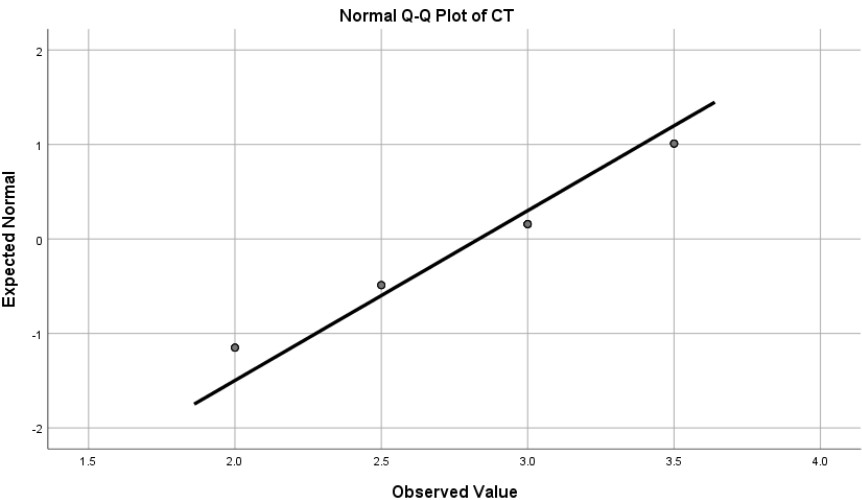

**Figure 3.** Normality test of CT.

*7.10. Focus Groups*

The focus group questions covered three areas: The first area, on which the teachers were placed, was the ease-of-use of the application (Ease of Access, Response Speed, Stability, Aesthetics, Technical support, Compatibility with school procedures). The second area was the usefulness of the application (as an evaluation tool, as a scorecard, as a material monitoring tool, as a feedback tool). The third area was the level of satisfaction with the application (Effectiveness, Compatibility with distance learning) and the integration of computational thinking criterion into the evaluative process.

The discussion was moderated by the researchers, and there was also an observer who signed the proceedings. During the process, notebooks and evaluation sheets with the areas and criteria that had been selected were distributed. With the guidance of the researchers, the discussion and recording began. After all issues had been investigated,

the recording ended. The procedure did not exceed two hours in each of the two groups. The duration of a focus group must be 1–2 h, based on the number of questions, the number of participants and the complexity of the topic [53]. In order to analyse the focus group data, the researcher identifies keywords used by respondents frequently as indicators of important themes [53]. Thus, the materials used for the research and analysis were the transcripts of the discussions.

### 7.10.1. Teachers' Qualitative Assessment of the Ease-of-Use of the Application

Statements of teachers (N = 11) regarding "Ease of Access" included expressions such as "Easy access", "I didn't face any problem with the access", "I had no difficulty with the access", "I didn't encounter any problem with the access", "Very good access" and "Exceptional access". Statements of teachers (N = 9) regarding the "Response Speed" included expressions such as "Fast response speed", "I didn't have any problem with the speed", "Entries were saved normally" and "Immediate and effective entries". Two of the teachers reported that the application was slow to load. Statements of teachers (N = 10) regarding "Stability" included expressions such as "Everything was working fine", "It didn't crash", and "I didn't have any problem with the stability". One teacher reported that he failed to save data. Statements of teachers (N = 7) regarding "Aesthetics" included expressions such as"I liked it", "Satisfactory colors", "It wasn't bad" and"Good colors". Four of the teachers reported that the application needed to be improved in terms of color, because the colors were intense. Statements of teachers (N = 11) regarding "Technical support" included expressions such as "There was immediate response", "There was immediate help and support", "The problems were solved immediately" and "Excellent communication". Statements of teachers (N = 12) regarding "Compatibility with school procedures" included expressions such as "Totally compatible". According to the above, it can be concluded that the majority of teachers considered the Rubric Score Application to be easy to use.

### 7.10.2. Teachers' Qualitative Assessment of the Usefulness of the Application

Statements of teachers (N = 9) regarding "As an evaluation tool" included "Complete student's description without having to write", "General assessment of students' abilities", "Useful tool for student's assessment", "Reliable assessment result" and "Accurate tool". Three of the teachers reported that they should be able to add their own assessment criteria. Statements of teachers (N = 11) regarding "As a scorecard" included "Monitoring which students need help", "Monitoring where I should pay more attention", "The grades had absolute relevance to reality" and "Accurate tool in terms of grades". One teacher reported that he was not satisfied with the assessment process. Statements of teachers (N = 5) regarding "As a material monitoring tool" included 'It was useful'. Seven of the teachers reported that they did not use it. Statements of teachers (N = 10) regarding "As a feedback tool" included "I know if I have to give more homework to students", "I know if I have to readjust my teaching", "It helps me to communicate with parents", "I know what exercises to choose for repetition" and "I monitor the student's assessment frequency". Two of the teachers reported that they were not satisfied because they could not add comments for the students. According to the above, it can be concluded that the majority of teachers considered the Rubric Score Application to be useful.

### 7.10.3. Teachers' Qualitative Assessment of Their Satisfaction with the Application

Statements of teachers (N = 12) regarding "Effectiveness" included expressions, for example, "Adequate tool", "Suitable for this purpose", "Satisfactory". Statements of teachers (N = 12) regarding "Distance learning" included expressions, for example, "Perfectly compatible", "Requires less time". According to the above, it can be concluded that the majority of teachers considered the application effective and satisfactory. This is a very important finding because teachers' usefulness perception encourages them in their personal development and improves their work in the classroom [38].

### 7.10.4. Teachers' Qualitative Assessment of the Integration of Computational Thinking Criterion into Evaluative Process

Statements of teachers (N = 9) regarding "Computational thinking" included expressions such as "Basic criterion", "We have to take it into account", "Necessary criterion", "Useful criterion" and "Problem solving is very important". Two of the teachers reported that they did not understand the concept of computational thinking, and one of the teachers reported that it is not an objective criterion. Additionally, two of the teachers reported that the teacher must follow a specific methodology before assessing the specific criterion. According to the above, it can be concluded that the majority of teachers considered the computational thinking criterion to be necessary. The assessment aims to properly cultivate CT skills, and educators must understand students' needs and build their awareness of CT [16].

### 7.10.5. General Comments

Finally, the teachers commented that, before the assessment, they explained the criteria to the students so that they understood the assessment method and the concept of self-assessment in depth. They also stated that they would like to be able to add their own criteria to the application to allow them to assess students with special educational needs, to change the coefficients of each criterion and to rate on a multi-point scale. Regarding the criterion of computational thinking, they stated that the teacher should be trained on the concept of computational thinking and the methodology to be followed in order to integrate it into the educational process. This result validates the opinion that CT is the "connecting tissue" between disciplinary knowledge and computer science [7]. The second version of this application has been suitably improved based on the conclusions obtained from the research. Finally, the teachers reported that the students should be assessed with the application in all courses. It is suggested that the school principal should be the administrator of the application, and the teachers should integrate the qualitative assessment into their courses.

## 8. Discussion

The present study applied the TAM model to predict teachers' satisfaction in the context of student assessment using a Rubric Score Application. All rubrics consist of six evaluative criteria with Graded Criteria Scales. The computational thinking criterion was added to the rubrics, since computational thinking has been heralded as a fundamental skill for the 21st century. In contrast with most related studies, this criterion was applied to other subjects besides the computer science course. Science and engineering practices represent one of the three-dimensional science standards, according to the Next Generation Science Standards (NGSS). Ref. [54] contains the K–12 science content standards, which set the expectations for what students should know and be able to do. Technology and engineering literacy included in the ongoing assessment of what US students in K-12 classrooms know and can do, according to the National Assessment of Educational Progress (2018). These educational reforms highlight the need for students to be exposed to computational thinking in the K-12 curriculum [55]. The following paragraphs illustrate the results of the research questions.

The first research question was 'Is it probable that this Rubric Score Application will be perceived as useful to a significant extent by the teachers?'. After evaluating teachers' responses, the results revealed that it is probable that this Rubric Score Application will be perceived as useful to a significant extent by the teachers. Thus, it could positively influence teachers' attitudes and increase their intention to use the application, a conclusion that is in line with that of [36]. The responses of teachers to the focus group questions regarding the usefulness of the application (as an evaluation tool, as a scorecard, as a material monitoring tool, as a feedback tool) were positive. Three teachers reported that they should be able to add their own assessment criteria, one teacher reported that he was not satisfied with the assessment process, seven teachers reported that they did not use it as

a material monitoring tool and two teachers reported that they were not satisfied because they could not add comments for the students.

The second research question was 'Is it probable that this Rubric Score Application will be perceived as easy to use to a significant extent by the teachers?'. The results revealed that it is probable that this Rubric Score Application will be perceived as easy to use to a significant extent by the teachers. Thus, it could positively influence teachers' attitudes and increase their intention to use the application, a conclusion that is in line with that of [36]. The responses of teachers to the focus group questions regarding the ease-of-use of the application (ease of access, response speed, stability, aesthetics, technical support, compatibility with school procedures) were positive. Only two teachers reported that the application was slow to load, one teacher reported that he failed to save data and four teachers reported that the application needed to be improved in terms of color, because the colors were intense.

The third research question was 'Does the perceived ease-of-use of this Rubric Score Application have a significant effect on its perceived usefulness?'. The results allowed us to conclude that the perceived ease-of-use variable has a significant effect on the perceived usefulness. These results are in line with the approach of the TAM theory, which states stating that the usefulness variable is affected by the ease-of-use variable for the system [36].

The fourth research question was 'Do the perceived usefulness and perceived ease-of-use have a significant effect on teachers' satisfaction?'. The results allowed us to conclude that the perceived usefulness variable has a significant effect on the satisfaction variable. This result is in line with the approach of the TAM theory, which states that the satisfaction (attitude) variable is affected by the perceived usefulness variable [36]. The responses of teachers to the focus group questions regarding teachers' satisfaction (effectiveness, compatibility with distance learning) were positive. In contrast, the perceived ease-of-use variable was shown to have no significant effect on the satisfaction variable.

The fifth research question was 'Is there a significant difference between teachers who teach STEM and non-STEM courses in terms of the frequency of use of computational thinking as an assessment criterion?'. The results revealed no significant difference between STEM and non-STEM teachers in terms of the frequency of use of the computational thinking criterion. Most of the teachers were willing to incorporate it in the evaluation process, regardless of their subject area.

The sixth research question was 'What are the teachers' attitudes towards the computational thinking criterion?'. The results revealed that teachers used the criterion of computational thinking without facing great difficulties regarding their subjects. The responses of teachers to the focus group questions regarding computational thinking were positive. Two of the teachers reported that they did not understand the concept of computational thinking, one teacher reported that it is not an objective criterion and two teachers reported that the teacher must follow a specific methodology before assessing the specific criterion. Finally, all teachers agreed that they need to be trained in the concept of computational thinking and the methodology followed in it to incorporate it into the educational and assessment procedure. The last outcome can also be confirmed by recent research concerning Greek teachers' attitudes toward Computational Thinking, where the majority of participants differentiated in terms of what Computational Thinking consists of and how familiar they are with it. They all agreed that they needed more dedicated training [56].

## 9. Conclusions

It is obvious that technology is changing rapidly, forcing education systems all over the world to adapt their methods and approaches in line with the new challenges and demands. The use of CT in conjunction with a valid and comprehensive teaching and learning assessment method encourages both teachers and students to make qualitative and more dedicated progress in general. This is why the development of CT skills is very important, and this research aims to boost this area. Integrating assessment with instruction could increase student engagement and improve learning outcomes [15,17,57–59]. The present

study focused on a student assessment procedure by adopting the Descriptive Assessment approach and creating an application consisting of rubrics with the addition of one CT criterion. The Score Rubric Application was evaluated as an assessment tool, a scorecard, a material monitoring tool and a feedback tool. It was considered to be useful to a significant extent by teachers, a result that addresses the first research question. Its evaluation in terms of ease-of-access, response speed, stability, aesthetics, technical support, and compatibility with school procedures indicated that it was considered to be easy to use to a significant extent by teachers, a result that concerns the second research question. Thus, it can influence positively teachers' attitudes and increase their intention to use the application, a conclusion that is in line with the TAM theory approach. The results of the third research question indicated that the perceived ease-of-use of the application influences its perceived usefulness, a finding that is in line with the TAM theory approach. In answer to the fourth research question, the results revealed that the perceived usefulness of the application influences teachers' satisfaction relative to its effectiveness and its compatibility with distance education, a finding that is in line with the TAM theory approach. A comparison of the frequency of using computational thinking as an evaluation criterion by teachers who teach STEM versus non-STEM subjects showed no significant difference. Additionally, teachers stated that they use computational thinking as an assessment criterion without facing great difficulties in their subject areas. The above results concern the fifth and sixth research questions. Finally, no significant difference was found when comparing the quantitative and qualitative evaluation results. It is worth noting that the majority of teachers used computational thinking as an assessment criterion and stated that they did not face any significant problem with it. However, at the end of the focus group questions, they concluded that they consider it to be necessary to participate in a training program regarding the concept and principles of computational thinking and the way they could integrate it into the educational process. As far as the research limitations are concerned, it is true that the pandemic situation discouraged teachers from participating in the research phase, thus creating a barrier to the generalisation of the research results. Future research should increase the number of variables and test their influences and inclusion in the TAM method. The purpose will be to contribute to the improvement of the performance of the application that will be used by teachers.

**Author Contributions:** Conceptualisation, I.D., K.L. and C.V.; methodology, I.D. and K.L.; software, I.D. and A.K.; validation, C.V. and A.K.; formal analysis, C.V. and I.D.; investigation, I.D., K.L., C.V., A.K. and A.A.; resources, I.D, C.V. and K.L.; data curation, I.D, C.V., K.L. and A.K.; writing—original draft preparation, I.D., C.V.; writing—review and editing, A.K. and A.A.; visualisation, I.D. and C.V.; supervision, A.K. and A.A. All authors have read and agreed to the published version of the manuscript.

**Funding:** Chrysoula Velaora is a recipient of financial support in the context of a doctoral thesis. The implementation of the doctoral thesis was co-financed by Greece and the European Union (European Social Fund-ESF) through the Operational Programme—Human Resources Development, Education and Lifelong Learning—in the context of the Act—Enhancing Human Resources Research Potential by undertaking a Doctoral Research" Sub-action 2: IKY Scholarship Programme for PhD candidates from Greek Universities.

**Institutional Review Board Statement:** Not applicable.

**Informed Consent Statement:** Informed consent was obtained from all subjects involved in the study.

**Data Availability Statement:** Not applicable.

**Conflicts of Interest:** The authors declare no conflict of interest. The funders had no role in the design of the study; in the collection, analyses, or interpretation of data; in the writing of the manuscript; or in the decision to publish the results.

## Appendix A

**Table A1.** Rubric.

| Criterion | Performance Level | | | |
|---|---|---|---|---|
| | **1** | **2** | **3** | **4** |
| **Reading comprehension** | Fails to extract and process primary and secondary information from texts; identifies parts of speech and their functions. | With significant difficulty and help, extracts and processes primary and secondary information from texts; identifies parts of speech and their functions. | Satisfactorily extracts and processes primary and secondary information from texts; identifies parts of speech and their functions. | Extremely easily extracts and processes primary and secondary information from texts; identifies parts of speech and their functions. |
| **Writing** | Writes with poor ideas, many lexical and grammatical errors, poor vocabulary and stereotypical style. | Writes with several lexical and grammatical errors, moderate vocabulary and often stereotypical style. | Writes with few lexical and grammatical errors, satisfactory vocabulary and sometimes personal and sometimes stereotypical style. | Writes extremely easily without lexical and grammatical errors, with a wide vocabulary, various communicative goals and personal style. |
| **Critical thinking** | Fails to combine information, explain reasoning, and argue about situations asked of them (hypothetical or real) | With significant difficulty and help, combines information, explains reasoning, and argues about situations asked of them (hypothetical or real) | Satisfactorily combines information, explains reasoning and argues about situations asked of them (hypothetical or real) | Extremely easily combines information, explains reasoning, and argues about situations asked of them (hypothetical or real) |
| **Participation-collaboration** | Does not participate in collective actions and hesitates to ask for and offer help and does not seem to appreciate the value of the group. | Rarely participates in collective actions and often hesitates to ask for and offer help and does not have a high appreciation of the value of the group. | Several times participates in collective actions and often asks for and offers help and appreciates the value of the group. | Actively participates in collective actions and does not hesitate to ask for and offer help and appreciates the value of the group. |
| **Diligence** | Does not complete exercises or tasks assigned to them in the school environment or at home. | Rarely completes exercises or tasks assigned to them in the school environment or at home. | Often completes exercises or tasks assigned to them in the school environment or at home. | Always completes exercises or tasks assigned to them in the school environment or at home. |
| **Computational thinking** | Never uses flexible methods that improve learning and/or solve teaching or everyday problems. | Rarely uses flexible methods that improve learning and/or solve teaching or everyday problems. | Often perceives and implements more effective methods for solving teaching or everyday problems. | Always analyses, documents and implements flexible methods of solving teaching problems or everyday problems. |

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
