# Peer review of "How a Rubric Score Application Empowers Teachers’ Attitudes over Computational Thinking Leverage"

_information, doi:10.3390/info14020118_

Round 1
Reviewer 1 Report
Thank you for the opportunity to review this manuscript, which I found to be generally well-written and interesting. The application of rubrics within the framework of Conceptual Thinking is indeed interesting. Below, I present a number of suggestions, which aim at strengthening the article further.
Abstract
The Abstract is clear in terms of scope and findings. However, you can add a sentence informing readers on the methods used (Questionnaire & Focus Groups), along with the sample.
Introduction
A clear purpose is discussed, along with relevant theoretical grounds. On the last paragraph of the Introduction, at the end, please add another sentence, explaining the structure of the paper. This would be beneficial to the reader, who will be able to formulate a clearer understanding in terms of organization and navigation within the text.
Conceptualization
While the theoretical analysis is interesting, I was expecting to see a wider range of updated sources – both in the introduction and Literature Review. I appreciate the importance of classic works published in the 1980s and 1990s. However, it’s important to include material that has been published in the last 5 years.
Moreover, I would advise you to use the following source in sections relating to quality teaching and measurement:
Efthymiou, L., Zarifis, A., & Orphanidou, Y. (2021). A Measurement Model for Collaborative Online Learning in Postgraduate Engineering Management Studies. In D. Ktoridou (Ed.), Cases on Engineering Management Education in Practice (pp. 1-21). IGI Global. https://doi.org/10.4018/978-1-7998-4063-3.ch001
Methods
The heading ‘Data Analysis’ is quite confusing. I would suggest renaming it to ‘Material and Methods’, as suggested in the Journal’s ‘Instructions for Authors’
Results
The analysis of results is interesting. I wonder whether you could link some of the findings back to the literature.
Conclusion
In the discussion, perhaps you can make a stronger case for the original contributions the paper makes. In other words, why is this study needed now and how does it advance our understanding of relevant theoretical or empirical matters? Also, a second point concerns the limitations of the study. While you make a brief reference to it, it’s important to explain the impact of the limited experimental sample on generalization – both in the conclusion as well as in the Methods’ section in page 5.
Very good work overall. I look forward to receiving a revised version of the paper.
Cordially,
Author Response
Abstract
In the middle of the abstract, I added the below sentence: A participatory of 15 teachers (n=15) were invited to a focus group discussion in which they were asked to complete a questionnaire and subsequently to openly analyze their answers.
Introduction
At the end of the last paragraph I added: At the end of the teachers’ assessment period, researchers analyzed the collected data and concluded specific findings. Researchers also organized a participatory focus group and in conjunction with the assessment data formulated the research results.
Conceptualization
At the end of the Related Work section, I added the below sentence: Moreover, as far as the online learning assessment is concerned, a measurement model based on metrics in combination with techniques such as, ‘peer interaction’, ‘forum activities’, ‘learning by doing’, and ‘systematic feedback’ is used. This model enables teachers to measure students’ performance and develop a more adaptive teaching approach [35].
The reference is for:
Efthymiou, L., Zarifis, A., & Orphanidou, Y. (2021). A Measurement Model for Collaborative Online Learning in Postgraduate Engineering Management Studies. In D. Ktoridou (Ed.), Cases on Engineering Management Education in Practice (pp. 1-21). IGI Global. https://doi.org/10.4018/978-1-7998-4063-3.ch001
I also included material which has been published in the last 5 years. You can find them in the References section with numbers : [7],[15-17],[19],[35],[38],[58,59]
Methods
I substituted the 'Data Analysis' with 'Material and Methods'
Results
7.10.3 At the end of the paragraph, I added the below text: This is a very important finding because teachers’ usefulness perception encourages them in their personal development and improves their work in the classroom [38] with direct mention to the literature [38]
7.10.4 At the end of the paragraph, I added the below text: Assessment aims to properly cultivate CT skills and educators must understand students' needs and build awareness of CT with direct mention to the literature [16]
7.10.5 In the middle of the paragraph, I added the below text: This result comes to validate the opinion that the CT is the “connecting tissue” between disciplinary knowledge and computer science with direct mention to the literature [7]
Discussion
1. At the beginning of the Conclusion section I added the below text: It is more than obvious that technology is changing rapidly forcing education systems all over the world to adapt their methods and approaches in line with the new challenges and demands. CT in conjunction with a valid and comprehensive teaching and learning assessment method encourages both teachers and students to make qualitative and more dedicated progress in general. This is why the development of CT skills is very important and this research aims to boost this direction.
2. At the end of the last paragraph of the Conclusion Section, I added the below text: As far as the research limitations are concerned, it is true that the pandemic situation discouraged teachers to participate in the research phase, thus causing a barrier to the generalization of the research results. Future research should increase the variables and test their influence and inclusion in the TAM method. Its purpose will be to contribute to the improvement and increase of the performance of the application that will be used by the teachers.
A similar text was added in the Material and Methods section.
Reviewer 2 Report
Dear Authors,
I have read with interest your manuscript which studies the role and importance of Computational Thinking (CT) in school curriculum educational activities and assessment process. The study proposes the use of Rubric Score Application and analyzes 6 research directions associated with CT. Your investigation includes a lot of analyzed data, is current and logically structured. You mentioned the limitations of the study, so I can offer some suggestions for improving the current version:
1. The citation style for multiple references would be useful to adapt to MDPI journals. For example Line 25 - [4], [5], line 41 - [10], [11], [12] etc. I think it is correct: [4,5], [10-12]. I think this journal article is useful as a template for references: https://www.mdpi.com/2078-2489/14/2/78
2. Reference 30 is quoted excessively (lines 108, 110, 530, 541, 551, 556). This reference is certainly important for your study, but it would be advisable to compare the results obtained with other similar scientific investigations.
3. Participants: lines 156-157. What was the sampling/selection criterion of the teachers? Could you specify the average age of the batch/group studied.
4. Line 4: STEM and non-STEM teachers........perhaps for some readers it would be useful to also indicate the expressions associated with the abbreviations (Science, Technology, Engineering and Mathematics).
5. Table 1/page 5 – (Teaching using CT's principles). It would be useful to add the non-responding teacher to the table to make 15 participants (you mentioned this / line 168).
6. Lines 163-165: Finally, 8 teachers declared that they have not at all or low experience in using ICT in student's assessment and 7 teachers that they have satisfactory or good experience. A comparison between these 2 categories would be interesting. Maybe you can use it in other publications.
7. Perhaps you can specify the software used in the statistical processing of the questionnaire data.
8. Paragraph 6.3. Data Analysis Methods: You have described in detail all the applied statistical tests, maybe you can summarize them. For example, testing the normality of the distribution of the data by the Kolmogorov-Smirnov variant could not be applied because you had few participants in the research. You only used Shapiro-Wilk. It is only a suggestion, which would reduce the size of the manuscript.
9. Factor 4 (Computational Thinking/CT – which includes Q10 and Q11) is missing from Table 3./Validity of items, Table 4./Reliability of factors and Table 5./Correlation of factors. These are the items to which you applied the 4-level Likert scale (1-4).
10. The formulated and tested hypotheses (H0-H11) would be better associated/joined with the 6 directions investigated by your study (lines 126-136).
Author Response
Response 1: The citation style for multiple references was adapted to MDPI journals.
Response 2: In addition to TAM, the Unified Theory of Acceptance and Use of Technology (UTAUT) is being used widely as a Technology acceptance model. Both of them are significantly similar, thus we did adopt only the TAM for our research. We didn't find any other similar scientific approach.
Response 3: The research sample was selected by convenience sampling due to availability at the given time and willingness to participate in the research. The teachers participated voluntarily after a relevant invitation. The age of the participants was not included in the research questionnaire and we cannot specify their average age.
Response 4: The expressions associated with the abbreviation STEM were added in the abstract.
Response 5: The non-responding teacher was added to table 1 (Teaching using CT's principles) to make 15 participants.
Response 6: Thank you for your advice. The comparison of these two categories would be really interesting. Maybe we will use it in other publications.
Response 7: The statistical package that was specified in Data Analysis Methods was the IBM SPSS 26.
Response 8: Reference to the Kolmogorov-Smirnov variant was removed.
Response 9: Computational Thinking/CT was added to Table 3./Validity of items, Table 4./Reliability of factors and Table 5./Correlation of factors.
Response 10: The formulated and tested hypotheses H0, H1, H2, H3, H10, H11 were better associated with the directions investigated by the study.